# Gender Differences in Migration †

**Francesca Ena**

Ambulatorio Medicina delle Migrazioni, ASL Gallura, 07026 Olbia, Italy; francesca.ena@aslgallura.it
† Proceedings from "Gender differences in diabetes" held in Olbia, 4–5 December 2020.

**Abstract:** There are about 200 million people on the move in the world, and approximately 50% of them are women. There are no clear migration plans for women leaving as a result of persecution, war, famine, climatic disasters or moving away from contexts of external abuse and even intrafamily violence. Gender-related violence, to which women are exposed in cultural contexts characterized by a patriarchal social organization, is manifested through different ways including, but not limited to, early marriages and genital mutilation, with reproductive health already being seriously impaired at an early age. To this must be added the consideration that low-income countries are not able to deal with chronic degenerative diseases with a multidisciplinary approach such as diabetes. Fragile or non-existent health systems are not prepared for this need, which now affects a third of all deaths from this cause. Compared to Italian mothers, women from high-migration pressure countries had a higher risk of gestational diabetes; in addition, young women of Ethiopian ethnicity are more exposed to increased diabetes risk, in an age- and BMI-dependent way. Gender inequalities are also more evident in migrants for other non-communicable diseases besides diabetes. A major effort is needed in terms of training practitioners and reorganization of basic health services, making them competent in an intercultural sense. Health education of the population as a whole and of women specifically is also needed to contain risk behavior and prevent the early onset of metabolic syndromes in general and of type 2 diabetes in particular.

**Keywords:** migration; low income; diabetes; gender

## 1. Introduction

According to the United Nations International Highlights Report of 2017 [1], there are about 200 million people on the move in the world. Of these, approximately 50% are women. Europe, in particular, is a destination for women who migrate alone, so-called breadwinners, or with a family member, following a plan totally or only partially shared. This migration plan is particularly weak or nonexistent at all for women who leave as a result of persecution, war, famine, climatic disasters or move away from contexts of external abuse and even intrafamily violence. Gender-based violence is heavily present not only in families, but also in social, employment and institutional contexts. We might consider this as a sort of feminization of poverty, becoming one of the strongest push factors that force many young women to leave their country, seeking salvation in completely different contexts. According to the sociologist Mara Tognetti in her review, "*migration is a way to escape cultural references and lifestyles no longer shared*" [2].

As topics related to gender differences in migration are numerous and complex to develop, following the presentation carried out at the meeting "Gender differences in diabetes" (Olbia, 4–5 December 2020), I will mainly address two aspects: (1) motherhood and (2) the greater possibility of the migrant developing diabetes mellitus once they arrive in the host country.

## 2. Motherhood as a Specific Gender Difference in Migration

Gender-related violence, to which women are exposed in these cultural contexts characterized by a patriarchal social organization, is manifested through labor exploitation,

leaving education early, early marriages and even genital mutilation. As a result of this last factor, reproductive health is often already seriously impaired at a very young age. The phenomenon of early marriages is widespread in many countries but reaches significant levels in Central and West Africa, from Chad, where it reaches peaks of 30% in girls under the age of 15, and in Niger, where 70% of girls under the age of 18 go to early marriages, according to the UNICEF report "Achieving a future without child marriage" [3]. Early marriages correlate with school dropout and poor empowerment of young women. Early pregnancies are a direct consequence of this, helping to increase maternal mortality rates and negative outcomes for infants. The practice of female genital mutilation (FGM), spread throughout the equatorial belt and the Horn of Africa, is an emblem of gender violence. It damages the reproductive health of women to different extents, depending on the type to which they are subjected, and interferes heavily with the possibility of living sexually free from physical and psychological conditioning [4,5]. Among the medium- and long-term complications of FGM, obstetric fistula is of note [6,7]. Ischemia from the compression that the head of the fetus exerts during passage in the birth canal, in conditions of protracted dystonia and in the absence of timely obstetric assistance, generates the fistulous tract. The coexistence of some forms of FGM further aggravates the picture, often leading to the death of the fetus. This abnormal pathway allows the passage of feces and/or urine via the vaginal route, resulting in recurrent urinary infections that sometimes result in fatal renal insufficiency. The fate of these women is almost always marked by being victims of social and physical isolation on the part of the family and the community. According to the most recent WHO data, about two million women in sub-Saharan Africa live with this condition and, to this number, about 50–100 thousand cases every year must be added. Gender-based violence continues to undermine the physical and mental health of women, even during migratory routes, at the hands of family members, carers, traffickers, police forces and, unfortunately, even in landing countries. The Global Report on Trafficking in Persons (UNODC 2016) reports that over 75% of people who are victims of trafficking are women, and of these, 20% are minors with a constant global increase in recent years [8]. A large proportion (some 80%) of women victims of trafficking intercepted in Italy come from Nigeria, after having stopped in Libya, a country in which they have suffered further forms of physical and sexual violence, exposed to unwanted pregnancies, induced abortions and sexually transmitted diseases [9,10]. According to the 2018 Action Aid Report—Connected Worlds—the most significant push factor towards Europe, in the case of Nigerian girls, is gender violence.

For some of these women, the degree of awareness with respect to the purpose of migration is clear, but unfortunately, the capacity of aging and empowerment of most of them is almost nil. This inability is also the result of some rituals, such as voodoo practices, to which they are subjected in the country of origin, with the threat of serious consequences on their physical and mental safety as well as that of their family. These observations, although not supported by bibliographical references, originate from the structured interviews conducted with migrant women with an Intercultural Mediator that have made it possible to understand these mechanisms.

Starting street prostitution, with all the risks associated with it, almost always represents the wall against which the migratory plans of the female victims of trafficking are shattered. In this context, another act of violence against these victims is performed: the institutional violence on the part of the host countries, often too concerned about the protection of their borders rather than recognition of the precarious situation in which migrant women are, in particular when they are victims of trafficking. The lack of cultural skills and the condition of isolation that they undergo make it difficult for them to access and correctly use the health services of the host country. This also happens in Italy in the face of a body of legislation that guarantees essential levels of assistance to people in conditions of legal irregularity. The amendment of restrictive provisions such as the Italian Law 132/18 (c.d. Decreto Salvini) has made it more difficult to take charge of and integrate asylum seekers, as pointed out by the Council of Europe: *"the new measures do*

*not offer adequate guarantees to vulnerable persons, such as victims of abuse and torture".* These restrictive rules, which put the health of already fragile people at risk, only recently have undergone a kind of mitigation, thanks to the advocacy activity of several agencies and scientific societies that, for years, have been dealing with migrant health, in full awareness that poor standards lead to poor health for migrant communities.

The June 2018 editorial of Lancet Public Health "No public health without migrant health" reports what was discussed in Edinburgh, where 700 public health experts from 50 countries produced a statement aimed at soliciting international agencies, governments and public opinion in order to produce different policies about migrant populations and their role in determining their health profile.

Gender, understood not only as genetic sex but as a cultural interpretation of it, is a non-negotiable health determinant in the approach to assessing the health profile of migrant women. From the analysis of Birth Assistance Certificates—Cedap 2016—there are clear differences in access to health services for foreign women, in which the access time to these services is longer than for Italian women, thus reducing the possibility of implementing the now consolidated preventive measures. The Health Observance Report of 2019 confirms that pregnancy and childbirth represent moments of greater contact of foreign women with our health system. In recent years, rates of access have improved in the times recommended by the protocols for taking proper care during pregnancy, as well as the outcomes of newborn children of foreign mothers. However, the persistence of a significant gap between the north and south of the country, with regard to maternal mortality outcomes, cannot be ignored and both neonatal and infantile mortality outcomes follow the historical gap existing in Italy, in terms of health care for the population in general, from the north to the south. The Magazine of the Italian Society of Paediatrics in the article "The unequal Italy begins in cradle" reports alarming data, in particular for the children of women without documents, from sub-Saharan Africa, who have a risk of infant mortality four times higher—8.2/1000—compared to children born from an Italian mother living in the north [11].

Another gender-related issue is the use of voluntary termination of pregnancy, often repeated, by foreign women. Law 194, which regulates access to voluntary termination of pregnancy in Italy, requires that a report be drawn up annually to be submitted to Parliament. From the latest 2020 report, relating to the 2018 data, it is clear that in the face of a trend of continuous decline for Italian and foreign women, voluntary abortion rates persist significantly higher for the latter. It is certain that behind many of these voluntary pregnancy interruptions, there are "hidden" women victims of trafficking. In this complex context, the training of gender-oriented practitioners and the presence of cultural mediators is essential, which, in agreement with each other, can intercept the unclearly expressed needs of foreign women accessing services. The north–south gap in terms of morbidity and fetal maternal mortality correlates with the presence of qualified cultural mediators in the consultations, which are clearly more represented in the north.

In order to improve the quality of access to and proper use of health services by migrant, resident or transiting populations, a number of operational manuals have been published such as guidelines from international and national institutions (OIM-UNHCR, Ministry of Home Affairs, NGO-SIMM-INMP). These guidelines are intended for use by health care professionals, for various reasons, coming into contact with migrant women from the time of landing and on the reception route. Guidelines are very useful for the early detection not only of the disease related to exposure in the countries of origin or acquired in the host country but also for the early detection of women victims of trafficking or FGM. Among the guidelines that are particularly important are those related to the management of victims of intentional physical and/or sexual violence, representing the psychological suffering emerging as pathology among asylum seekers upon reception and, in particular, women exposed to gender violence.

### 3. Diabetes as a Non-Communicable Disease (NCD) Involved in Migration

In addition to problems related to reproductive health, we cannot ignore that the migrant population are faced with another additional health problem. In fact, especially for those residing in the host country for several years, a so-called epidemiological transition takes place, whereby the health system's focus should be on non-communicable diseases rather than on infectious and imported diseases. Diabetes is a critical issue in the so-called developing countries, where, given the growing population, it is one of the health emergencies from now and in the coming decades, as reported in "Non Communicable Diseases: A Priority for Women's Health and Development" [12]. This increase is particularly significant in the Middle East, North Africa, and sub-Saharan Africa followed by Southeast Asia. The genetic predisposition of some ethnicities and, above all, the forced urbanization with consequent modification of the alimentary habits and acquisition of risk lifestyles (smoke, alcohol, inadequate feeding) are the main reasons for the significant increase in type 2 diabetes in these populations. In this regard, the research carried out on the population of Ovahimba in the North of Namibia is interesting [13]. The authors found that alterations in cortisol homeostasis may link changes in sociocultural environment to increased diabetes and metabolic risk. This work highlights an increase in cortisol secretion related to stress from urbanization, resulting in a diabetogenic effect. In this population, urbanization is associated with an increasing prevalence of disorders of glucose metabolism and other unfavorable metabolic parameters. Besides changes in lifestyle, this may be attributed to an increased cortisol exposure due to living in an urban environment. In another population in Tanzania, an important shift toward an inflammatory phenotype has been associated with an urban lifestyle, providing data on the metabolic factors that may affect diabetes epidemiology in sub-Saharan African countries [14]. It should be taken into account that low-income countries are not able to deal with chronic degenerative diseases with a multidisciplinary approach. Fragile or nonexistent health systems are not prepared for this need, which now affects a third of all deaths from these causes [15,16]. Until now, health care in low-income countries has been based on the support of international donors, sometimes conditional, but focused on vertical interventions for the treatment and prevention of individual infectious diseases (for example, among many, the fight against HIV, tuberculosis, malaria, etc.). Such an approach cannot be transferred to the management of chronic degenerative diseases that cannot be managed with the care of single instants of their natural history. For some years, other intervention models, defined as diagonal, have been tested [17]. These models are capable of implementing strategies identifying priority areas of intervention, which may lead to general improvements in health systems such as the training of local human and professional resources, planning of the use of financial resources and infrastructure, the provision of medicines, and quality of services. This means that interventions intended for a single disease can also have positive effects on preventing and treating other forms of pathology as well as improving the quality of local health systems by overcoming sectoral interventions.

As far as Italy is concerned, the demographic trend towards an ageing population does not spare the migrant population [18]. The migratory phenomenon in Italy is now consolidated if we consider that this migratory balance was reversed in 1973. Currently, in Italy, people arrive at a young age and through their genetic profile, their exposure to strenuous work and unhealthy lifestyles (alcohol, smoking, inadequate nutrition, sedentary lifestyle) makes them prone to chronic degenerative diseases, among which type 2 diabetes plays a prevalent role, sometimes appearing even earlier than in the native population. In our country, the acquisition of Western lifestyles and scarce economic availability mean that, in many families, food of little nutritional value but rich in fat is consumed. The tendency to overweight and obesity in children and unhealthy behaviors such as poor physical activity are now frequent in these subjects. The consumption of sugary drinks does not spare children from migrant families, as shown by the Okkio national health surveillance system of 2016, coordinated by the Higher Institute of Health, where it was reported that in contrast to the observed decrease in childhood overweight and obesity

observed in Italian children from 2008 to 2016, this did not apply to migrant children [19]. Compared to Italian mothers, women from high migration pressure countries had a higher risk of gestational diabetes and of all considered adverse events [20]. This is also true for Ethiopian ethnicity, which in Israel, was found to be associated with increased risk of diabetes, in an age- and BMI-dependent way, since young Ethiopians (<50 yrs), particularly women, had the greatest increase in risk [21]. Recently, French authors feared that migrants shared an increased risk of uncontrolled diabetes and, therefore, migration could be a risk factor of uncontrolled diabetes. Knowing the migration history of migrant patients is fundamental to understand certain barriers of care [22]. It has been found that among both Greek-born and immigrant groups, women report substantially higher rates of NCDs, although gender inequalities are more pronounced among immigrants [23].

Human suffering, as a result of natural disasters or conflict, encompasses death and disability from NCD, including diabetes, which have largely been neglected in humanitarian crises [24]. Examining the evidence on the burden of diabetes, use of health services, and access to care for people with diabetes among populations affected by humanitarian crises in low-income and middle-income countries, as well as identifying research gaps for future studies are demanding tasks. The burden of diabetes in humanitarian settings is not being captured, clinical guidance is insufficient, and diabetes is not being adequately addressed. Crisis-affected populations with diabetes face enormous constraints accessing care, mainly because of high medical costs. Further research is needed to characterize the epidemiology of diabetes in humanitarian settings and to develop simplified, cost-effective models of care to improve the delivery of diabetes care during humanitarian crises [24,25].

Since foot lesions can originate from a solitary traumatic event and rapidly progress to gangrene and sepsis, loss of life from diabetic foot infections is both certain and uncounted. Moreover, the problem of diabetic foot disease might be magnified in Africa. The International Diabetes Federation estimates that 70% of Africans with diabetes are undiagnosed. Without a diagnosis, practices known to prevent or mitigate diabetic foot complications will not be implemented and people will die before presentation; this aspect might be amplified by migration, particularly irregular migration [26].

**4. Conclusions**

A major effort is needed in terms of training practitioners and reorganization of basic health services, making them competent in an intercultural sense. Health education of the migrant population as a whole and of women in particular is needed to contain risk behavior and prevent the early onset of metabolic syndromes in general and of Type 2 diabetes in particular [27]. This also applies to the protection of motherhood in migrant populations.

**Funding:** This research received no external funding.

**Conflicts of Interest:** The author declares no conflict of interest.

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
