# Peer review of "Gender Differences in Migration"

_diabetology, doi:10.3390/diabetology3020023_

Round 1

Reviewer 1 Report

In this submission from Ena F, the author tackles an important and challenging topic, the risk of chronic disease and lack of effective resources to treat it among the female migratory population. The author should be commended for addressing a critically relevant subject in today’s world, one that is often forgotten or overlooked. Unfortunately, while the concept of this manuscript is sound, it suffers from a lack of consistent organization and the importance of the authors message is lost on the reader. Introduction There are portions of the first two paragraphs that are direct duplicates of the abstract. The reader is left without an idea of the relationship of diabetes to the main thrust of the article. There is a general lack of organization within the introduction, leaving the reader searching for consistent themes in each paragraph. The section on diabetes risk in the migratory population occurs half-way through the article. It would improve the overall manuscript if it were woven through the entire body. While there is excessive discussion on a variety of health-care risks and injustices that occur within migratory women in the first half of the manuscript, the author does not make a convincing argument about the greater association of diabetes among migratory women compared to men. The author begins to discuss the inequalities between migratory population as compared to native populations within Italy, but this topic is underdeveloped. The authors would be well suited to explore this concept further, gathering important existing data in different European countries. This would strengthen the manuscript submission. The conclusion essentially reiterates the final sentence of the abstract.

Author Response

Thank you for your comments on the manuscript to improve its quality
I agree with the Referee that the topic of gender difficulties in migration and the risk of chronic complications due to lack of resources is a very important topic. This manuscript, however, does not intend to deal exhaustively with the subject, but to give an idea of its complexity and it is born, as I was asked by the guest editor of the special issue "gender difference in diabetes" on Diabetology, as proceeding of a conference on Gender Medicine held in Olbia in December 2020.

Having said that, I agree with the referee that the work was unorganized and for this reason the work was resumed completely. In particular, it is now clearly stated (final part of the introduction) that only two migration problems will be addressed: motherhood and the acquisition of diabetes, which constitute two separate chapters. I agree that a more global view of the problem would be a qualitative improvement, but for practical reasons, I refer to our Italian reality, even if some examples of other countries were already foreshadowed in the work.  In this revised version I did my best to improve the language and make the concepts clearer. Two new references (17 and 19 of the revised version) have been added.  I apologize again with the referee if the previous version turned out  sloppy. Thanking for the time on the referral process and for the comments, I hope that the manuscript is now considered acceptable.

Reviewer 2 Report

This commentary represents a laudable effort and provides valuable insight to crucially important subject. However, a lot of editing work is required to achieve it’s full potential. There is an overall tendency to poor sentence construction with rather long sentences ( e.g. 176 - 180). This made me have to read certain areas over and over again to get the message, making it quite a challenging read at times. That said, the content makes it clear that the author is passionate and knowledgeable about the subject matter.
Some specifics:

  • 75 - 79: I feel this needs to be referenced. All cultures have rituals and it’s important to provide evidence (even from qualitative interviews) of migrant women that these activities had a direct bearing on their situation. Otherwise this article runs the risk of being judged to have simply pandered to stereotypes whilst trying to address a very important subject.
  • 77: should be ‘voodoo’ not ‘wodoo’.

Author Response

I thank the Refere for the encouraging comments and for appreciating the purpose of the work. The manuscript has been completely rewritten in an attempt to avoid long and difficult sentences, the two specific points identified in lines 176-180 and 77 have been corrected. As regards the search for a bibliographical entry, point 75-79 unfortunately does not exist a bibliographical entry, but these considerations arose from interviews with migrant women with the help of the Intercultural Mediator. This has been added in the text. To better clarify the purpose of the work that arose as the proceeding of a congress on medicine gender differences  held in Olbia in December 2020, the purpose of the manuscript is now clearly indicated at the end of the introduction. Only two chapters of the migration problem will be addressed: motherhood and the acquisition of diabetes .  In this revised version I did my best to improve the language and make the concepts clearer. I apologize again with the Refere if the previous version turned out a little sloppy. Two new references (117 and 19 of the revised version) have been added. I hope the manuscript is now acceptable for publication , thanking again for the suggestions and comments.

Round 2

Reviewer 1 Report

The authors have made a number of organizational and syntax changes to the maunscript that have improved it to the point of consideration of publication.  This is a recap of a meeting proceedings (not previously designated that way), and it should be taken into account by editors.